# Household clustering and seasonal genetic variation of *Plasmodium falciparum* at the community-level in The Gambia

Marc-Antoine Guery[1], Sukai Ceesay[2], Sainabou Drammeh[2], Fatou K Jaiteh[2], Umberto D'Alessandro[2], Teun Bousema[3], David J Conway[4], Antoine Claessens[1,2,4]*

[1]LPHI, MIVEGEC, University of Montpellier, CNRS, INSERM, Montpellier, France; [2]Medical Research Council Unit The Gambia at the London School of Hygiene and Tropical Medicine, Banjul, Gambia; [3]Radboud University Medical Center, Radboud Institute for Health Sciences, Department of Medical Microbiology, Nijmegen, Netherlands; [4]Department of Infection Biology, London School of Hygiene and Tropical Medicine, London, United Kingdom

*For correspondence:
antoine.claessens@umontpellier.fr

## eLife Assessment

This article presents a **valuable** genetic spatio-temporal analysis of malaria-infected individuals from four villages in a highly seasonal transmission setting in The Gambia, covering the period between December 2014 and May 2017. Evidence generated by the study's laboratory and data processing approaches is **solid** and helps to advance the understanding of malaria in The Gambia, particularly due to its longitudinal design and the inclusion of asymptomatic cases.

**Abstract** Understanding the genetic diversity and transmission dynamics of *Plasmodium falciparum*, the causative agent of malaria, is crucial for effective control and elimination efforts. In some endemic regions, malaria is highly seasonal with no or little transmission during up to 8 mo, yet little is known about how seasonality affects the parasite population genetics. Here, we conducted a longitudinal study over 2.5 y on 1516 participants in the Upper River Region of The Gambia. With 425 *P. falciparum* genetic barcodes genotyped from asymptomatic infections, we developed an identity by descent (IBD) based pipeline and validated its accuracy against 199 parasite genomes sequenced from the same isolates. Genetic relatedness between isolates revealed a very low inbreeding level, suggesting continuous recombination among parasites rather than the dominance of specific strains. However, isolates from the same household were sixfold more likely to be genetically related compared to those from other villages, suggesting close transmission links within households. Seasonal variation also influenced parasite genetics, with most differentiation occurring during the transition from the low transmission season to the subsequent high transmission season. Yet chronic infections presented exceptions, including one individual who had a continuous infection by the same parasite genotype for at least 18 mo. Our findings highlight the burden of asymptomatic chronic malaria carriers and the importance of characterizing the parasite genetic population at the community-level. Most importantly, 'reactive' approaches for malaria elimination should not be limited to acute malaria cases but be broadened to households of asymptomatic carriers.

**eLife digest** Malaria is a life-threatening, mosquito-borne illness caused by the parasite *Plasmodium falciparum*. In 2023 alone, over 200 million people were diagnosed with the disease, and hundreds of thousands died from it. Yet the true scale of the problem is likely underestimated.

Indeed, most infected individuals do not experience symptoms and therefore do not seek treatment. However, they can still pass the parasite to any mosquito that bites them, thereby contributing to the disease silently persisting within the population.

These asymptomatic infections therefore complicate malaria eradication efforts, in part because they pose a challenge to studying *Plasmodium* genetics. Controlling the disease often requires a detailed understanding of how different strains of the parasite wax and wane in a population across regions and seasons; without considering asymptomatic individuals, this knowledge remains incomplete.

To address this issue, Guery et al. studied *Plasmodium* infections in four nearby villages in The Gambia for over two years. In this region, a long dry period during which malaria spreads slowly is followed by a short, wet season marked by high transmission levels. The team analysed samples from 425 asymptomatic participants, generating unique genetic 'barcodes' for each strain of the parasite they detected. These data were then used to construct *Plasmodium* 'family trees' and determine transmission dynamics.

The analyses revealed a high degree of genetic diversity in *Plasmodium* from this region, which increased during the wet season. This typically indicates high levels of transmission within the population, with frequent mosquito bites increasing the chance of *Plasmodium* strains recombining their genetic make-up. In contrast, *Plasmodium* parasites found in members of the same household were much more likely to be genetically related. This suggests that individuals living together were infected at the same time by one mosquito, or that the insects spread the disease between people living in close quarters. Finally, parasites could persist for over a year for certain asymptomatic individuals.

These results shed new light on the factors affecting *Plasmodium* spread and genetic diversity, highlighting the role of asymptomatic carriers and household-level transmission. Guery et al. hope their findings will pave the way to more effective malaria control strategies. Mass testing could be important to identify asymptomatic carriers, for example, as well as treating those living with infected individuals.

## Introduction

Malaria control and elimination efforts require a detailed understanding of *Plasmodium falciparum* population dynamics, particularly in regions with seasonal transmission. In The Gambia, substantial progress has been made in reducing malaria prevalence in the first decade of 2000. However, the decline has fallen short of meeting the 75% reduction target set for 2025, emphasizing the need for more targeted interventions (*World Health Organization, 2023*).

In regions characterized by seasonal transmission, malaria clinical cases peak towards the end of the wet season, a pattern mirrored in The Gambia's transmission cycle. Its climate is characterized by a rainy season from late June to September, followed by a ~8-mo-long dry season, typically without rainfall. Malaria cases are almost exclusively reported from September to December (the high transmission season), while malaria is virtually absent for the remaining 8 mo (the low transmission season) (*Ceesay et al., 2008*). During the high transmission season, malaria disproportionately affects rural, low-income populations in the eastern regions (*Ahmad et al., 2023*; *Mwesigwa et al., 2017*; *Mwesigwa et al., 2015*; *Sonko et al., 2014*). Understanding how the *Plasmodium falciparum* parasite population adapts to these seasonal variations is critical for informing targeted malaria elimination strategies.

Evidence suggests that imported cases play a minor role in maintaining malaria transmission, with the resurgence of malaria attributed to a persistent reservoir of asymptomatic chronic carriers that bridge two transmission seasons separated by several months of low to no transmission (*Daniels et al., 2015*; *Fola et al., 2023*; *Gwarinda et al., 2021*; *Sy et al., 2022*). Previous studies in The Gambia and neighboring Senegal demonstrated that individuals can harbor persistent infections across transmission seasons (*Ahmad et al., 2023*; *Collins et al., 2022*), with some infections maintaining transmissible

gametocytes (*Andolina et al., 2021*; *Barry et al., 2021*). This reservoir of asymptomatic infections represents a major challenge to malaria control programs.

Various molecular methods and metrics are now available to characterize the parasite genetic diversity. At the population level, the average COI—the number of distinct parasite genotypes within a host—is often used to infer transmission intensity (*Daniels et al., 2013*; *Hendry et al., 2021*; *Lee et al., 2021*; *Nkhoma et al., 2012*; *Pacheco et al., 2020*). Genetic relatedness is assessed through Whole Genome Sequencing (WGS) or the cost-effective molecular barcode genotyping, the latter able to identify a unique parasite strain with as few as 24 loci (*Daniels et al., 2008*). From pairwise distances between such barcodes, it has been shown that relatedness tends to decrease with time and distance of the sampled infections both at the national level in The Gambia and at the local level in neighboring Senegal (*Amambua-Ngwa et al., 2019*; *Lee et al., 2021*).

Population-level genetic analyses, particularly using IBD, provide a powerful tool for exploring parasite relatedness and recent recombination events. While Fst-based studies are valuable for long-term and large-scale comparisons, IBD allows for the resolution of recent recombination events, shedding light on fine-scale spatio-temporal transmission dynamics (*Mobegi et al., 2012*; *Noviyanti et al., 2020*; *Taylor et al., 2017*).

Extensive genomic epidemiology studies to date focused on clinical cases, yet such symptomatic cases only represent a minority of all *P. falciparum* infections (*Lindblade et al., 2013*; *Stone et al., 2015*). Thus, relying exclusively on clinical cases may lead to miss a substantial fraction of the actual parasite population diversity.

In this study, we analyzed *P. falciparum* genetic diversity in four nearby villages of The Gambia's Upper River Region through a longitudinal study spanning 2.5 y (*Collins et al., 2022*; *Fogang et al., 2024*).

Using a combination of molecular genotyping and whole genome sequencing, we constructed a high-quality pipeline to assess COI and genetic relatedness at the community-level. We aimed to elucidate how seasonal transmission cycles shape parasite population structure, including the stability of COI, the spatio-temporal patterns of IBD, the persistence of drug resistance alleles, and the contribution of asymptomatic chronic carriers to sustained transmission. These findings provide insights into the dynamics of parasite recombination and the genetic adaptations required for persistence in seasonal transmission settings.

## Results

### Combined barcode and whole genome analysis pipeline

Overall, 5322 fingerprick and 253 venous blood samples were collected from 1516 individuals aged 3–85 y in four nearby villages of the Upper River Region of The Gambia between December 2014 and May 2017, as detailed in previous studies (*Figure 1A*, *Supplementary files 1 and 2*; *Collins et al., 2022*; *Fogang et al., 2024*). To characterize the *P. falciparum* population genetic relatedness and identify the impact of antimalarial drugs, we attempted parasite genotyping on 442 isolates (*Supplementary file 3*; *Supplementary file 4*) and whole genome sequencing on 331 isolates (*Supplementary file 5*) for a combined total of 522 unique isolates over 16 time points (*Figure 1B*). Whole parasite genomes were successfully sequenced for 199 highly covered (between 3756 and 27516 high quality SNPs) isolates distributed across all 16 time points (*Figure 1—figure supplement 1*). Through the concatenation of SNPs from molecular barcode genotyping and whole genome sequencing, we obtained a high-quality 'consensus' barcode for 425 isolates comprising on average 65 SNPs (median of 64 SNPs, range 30–89 SNPs) (*Figure 1C*, *Figure 1—figure supplement 2*). A detailed description of the pipeline is available (*Figure 1—figure supplement 3*).

### Complexity of infection is stable across seasons

The complexity of infection (COI), defined as the number of unique genotypes/genomes within an infected individual, serves as an indicator of parasite strain diversity within the population. We estimated COI using $F_{ws}$ values from genomes and the proportion of heterozygous loci from genomes and barcodes (*Supplementary file 6*; *Supplementary file 7*). Across these metrics, the proportions of polygenotype isolates were estimated as 40% (ranging from 22 to 61% between time points) using $F_{ws}$ (80/199), 39% (range: 22 to 57%) using heterozygous loci of genomes (78/199), and 34% (range:

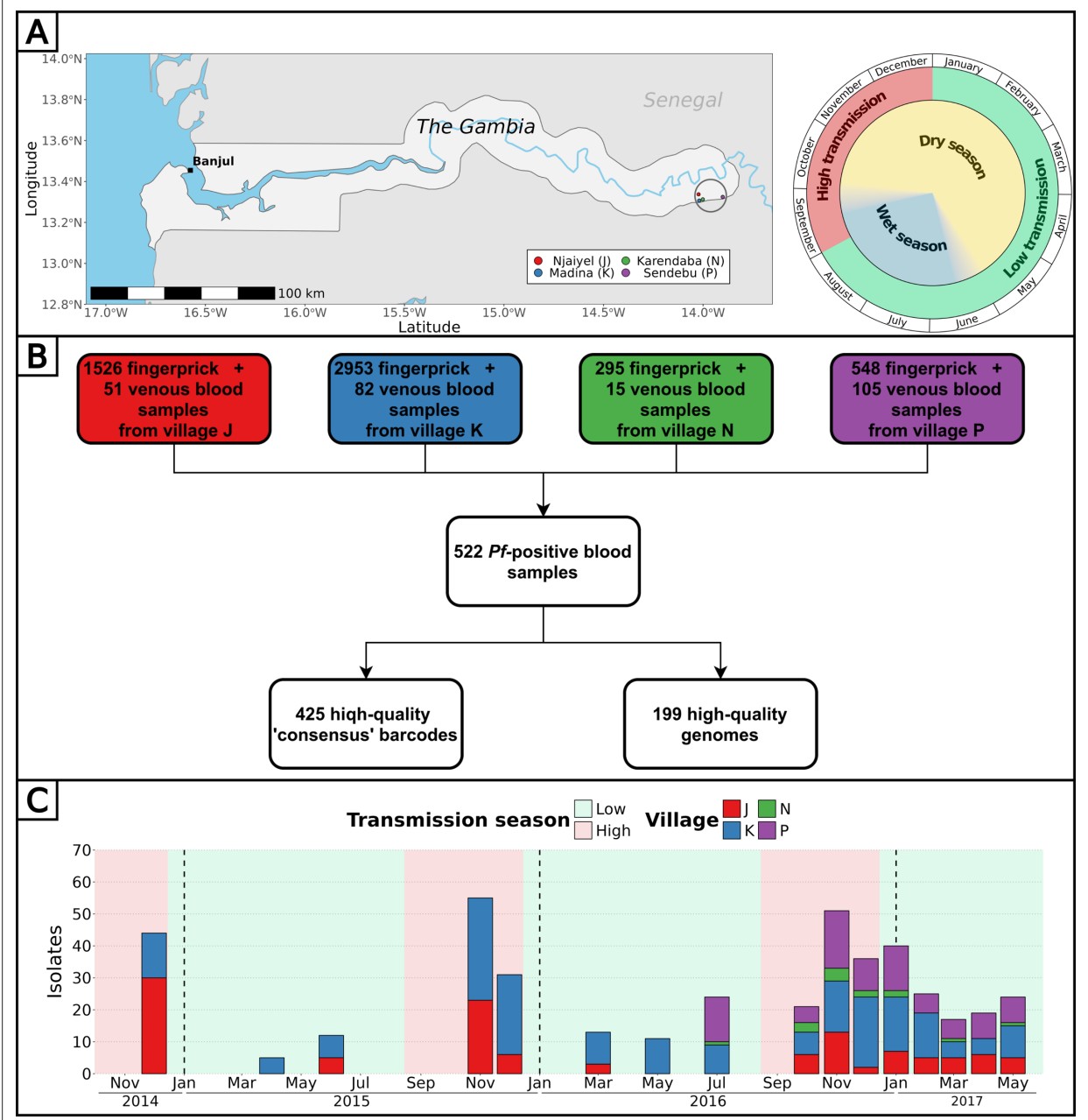

**Figure 1.** Study design and analysis pipeline. (**A**) Blood samples from all participants from four villages of the Upper River Region of The Gambia were collected up to 16 times over 2.5 y. The peak of clinical malaria cases occurs at the end and right after the rainy season. Made with Natural Earth. (**B**) Overall, 522 blood samples (307 fingerprick and 215 venous blood) were genotyped and/or whole genome sequenced, resulting in 425 high-quality barcodes and 199 high-quality genomes. Additionally, six drug resistance markers were successfully genotyped and/or called from whole genomes, in a total of 438 isolates. (**C**) High-quality barcodes and genomes were sampled in four villages over 16 time points between December 2014 and May 2017.

The online version of this article includes the following figure supplement(s) for figure 1:

**Figure supplement 1.** Number of high-quality barcodes (425 isolates) and genomes (199 isolates) successfully sequenced over 16 time points between December 2014 and May 2017.

**Figure supplement 2.** Number of homozygous single nucleotide polymorphisms (SNPs) per barcode.

**Figure supplement 3.** Steps of the combined barcode- and genome-analysis pipeline using 522 *P. falciparum*-positive blood isolates.

**Figure supplement 4.** Correlation between $F_{ws}$ values and percentage of heterozygous loci.

**Figure supplement 5.** Proportion of polyclonal isolates over time estimated by $F_{ws}$ ($F_{ws} < 0.95$) and the proportion of heterozygous loci (more than 0.5% of available sites) on barcodes and genomes.

*Figure 1 continued on next page*

Figure 1 continued

**Figure supplement 6.** Comparison of heterozygous locus calls between each pair of molecular barcode (obtained by genotyping) and their corresponding genomic barcode (built from allelic frequencies).

**Figure supplement 7.** Comparison of molecular barcodes loci (built from genotyped single nucleotide polymorphisms, SNPs) and consensus barcodes loci (combining molecular and whole genome sequencing, WGS loci) with WGS loci for two groups of samples clustered by their collection dates.

18 to 62%) using heterozygous loci of barcodes (145/425). As expected, the proportion of heterozygous loci showed a strong negative correlation with $F_{ws}$ both for barcodes ($R^2=0.83$, p-value $< 10^{-15}$) and genomes ($R^2=0.92$, p-value$< 10^{-15}$), indicating its efficacy as a predictor of COI (*Figure 1—figure supplement 4*). Although the proportion of polygenotype isolates fluctuated between time points, no discernible trend was observed between low and high transmission season, suggesting a relatively stable complexity of infections in the population throughout the 2.5 y study duration (*Figure 1— figure supplement 5*). At the individual level, we previously showed that the COI was stable in polyclonal infections during the dry season (*Collins et al., 2022*).

## Low *P. falciparum* inbreeding at the community-level

To determine the relatedness between isolates and compare malaria parasites from distinct geographical locations and distant times, IBD was calculated pairwise. The reliability of consensus barcodes in identifying genetically related isolates was confirmed through a strong linear correlation between barcode-IBD and genome-IBD, particularly when both were above 0.5 ($R^2=0.77$, p-value $<10^{-15}$), with this threshold chosen to distinguish between related (IBD ≥0.5) and unrelated (IBD <0.5) samples for all 425 consensus barcodes (*Figure 2—figure supplement 1*). The classifications (related or unrelated) resulting from barcode-IBD values and genome-IBD values (considered as the gold-standard) were in a strong agreement (Cohen's kappa of 0.839) with high values of specificity (0.997), sensitivity (0.841), and precision (0.843).

The level of inbreeding in the parasite population was characterized from relatedness estimated between barcodes sampled up to December 2016, discarding subsequent samplings in 2017 as they concerned exclusively a cohort of 74 asymptomatic carriers. To avoid sampling bias, only one barcode per continuous infection was retained, resulting in 284 unique barcodes sampled between December 2014 and December 2016. Among 34,326 pairwise comparisons of barcodes, only 435 (1.3%) demonstrated relatedness with an IBD above 0.5, indicating extensive outcrossing within the parasite population (*Figure 2A*). Out of the 284 barcodes, 73 (26%) were identical (IBD ≥0.9) to a barcode from another individual, while 140 (49%) were related (0.5≤IBD < 0.9) to at least one other barcode (*Figure 2B*). Although clusters of genetically related barcodes may suggest a higher connection within villages (1.6% of related barcodes) than between them (0.7% of related barcodes), the average proportion of related isolates was not significantly different (Welch's t-test value = 1.42, p-value = 0.24), indicating interconnectedness between villages.

## Pattern of relatedness between infections is shaped by seasonality

To explore the spatio-temporal relationship between barcodes from distinct individuals, we took advantage of our frequent samplings between December 2014 and December 2016 at the community-level to calculate the percentage of related isolates across six temporal groups, ranging from 0 to 2 mo apart to 16–24 mo apart between sample collections (*Figure 3A*). Remarkably, at the temporal level, the average proportion of related barcodes within barcodes sampled less than 2 mo apart is 4.7%. This value is 10-fold higher than the proportion of related barcodes sampled more than 12 mo apart equal to 0.30% (Welch's t-test value = 4.72, p-value $< 10^{-4}$). This indicates that recombination between *P. falciparum* isolates breaks down IBD with time.

Similarly, the pairwise IBD was computed for spatial groups: pairs of barcodes from the same household, different households within the same village, and different households across villages (*Figure 3B*). At the spatial level, barcodes sampled less than 2 mo apart and from the same household were six times more related than those sampled from different villages (average proportions of 0.093 and 0.015, Welch's t-test value = 3.30, p-value < 0.05). However, when barcodes were sampled more than 2 mo apart, the correlation between genetic relatedness and sampling location disappeared. Altogether, the overall low inbreeding combined with the increased proportion of related isolates

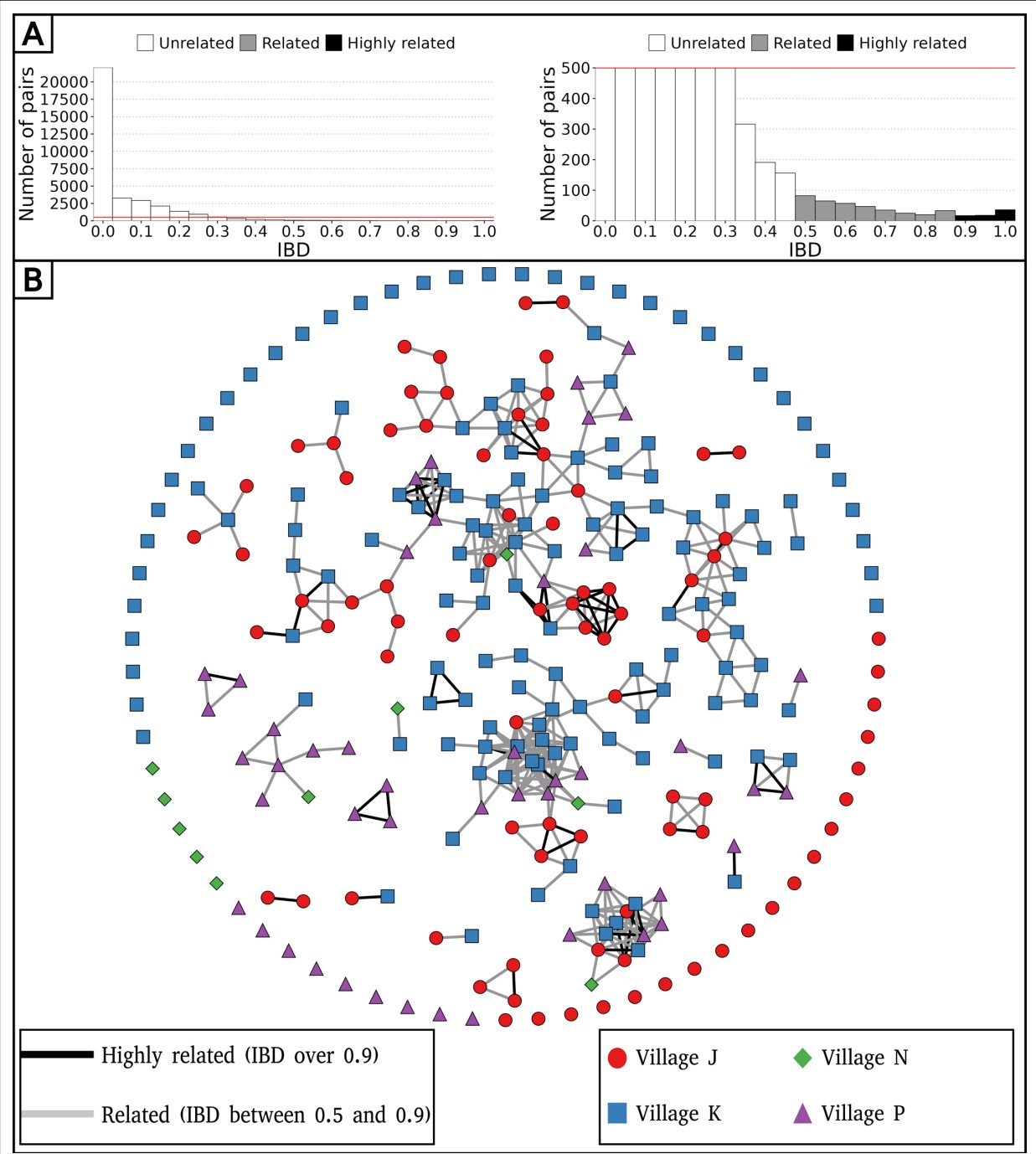

**Figure 2.** Low parasite inbreeding level in four villages in The Gambia inferred from inter-individual genetic relatedness. The genetic relatedness of parasites was assessed from barcodes sampled between December 2014 and December 2016, keeping just one barcode per continuous infection, which resulted in 284 remaining barcodes. (**A**) Distribution of identity by descent (IBD) values between barcodes (left panel), and with a cap at 500 pairs to highlight related barcodes at lower frequency (right panel). (**B**) Relatedness network of 284 isolates with barcodes represented as nodes and IBD values represented as edges. Barcodes are grouped into clusters using the compound spring embedder layout algorithm from Cytoscape (version 3.10.1).

The online version of this article includes the following figure supplement(s) for figure 2:

**Figure supplement 1.** Identity by descent (IBD) calculated from consensus barcodes highly correlates with IBD from genomes.

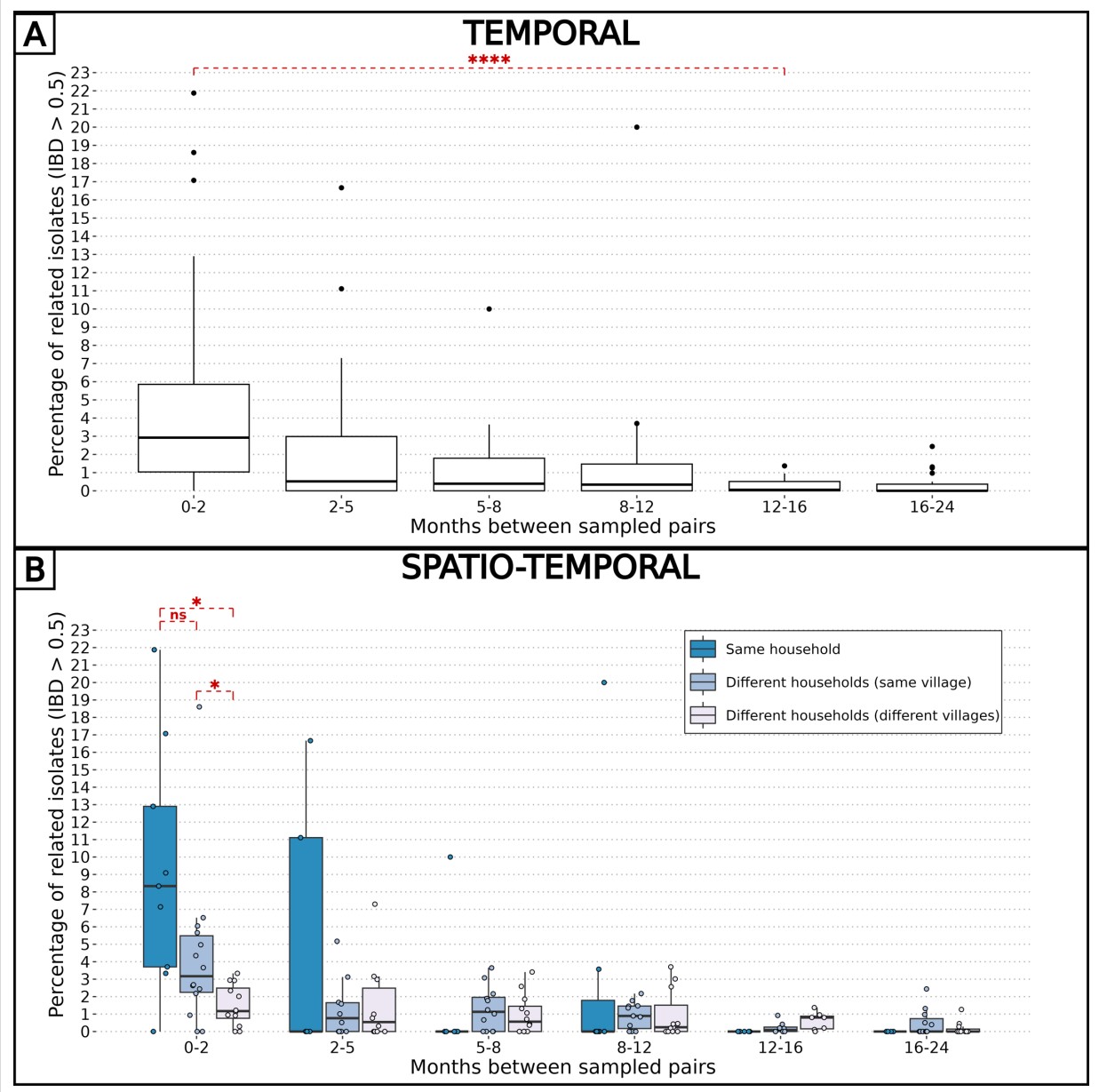

**Figure 3.** Combined effects of spatial and temporal distances on parasite-relatedness. The proportions of related barcodes (identity by descent, IBD ≥0.5) between each pair of households are binned into time intervals of various lengths such that the number of observations in each bin is similar. Box plots display the median, the first and third quartiles, and whiskers extending up to 1.5 times the interquartile range. Groups of related barcodes were compared with Welch's t-tests (*p-value < 0.05, ****p-value < 0.00005, ns: non-significant). (**A**) All pairs of isolates were grouped together in the same time interval. (**B**) Pairs of isolates were grouped by their relative spatial distance.

within the same household indicate a scenario in which the same infectious mosquito infected two or more individuals living together, or a direct transmission chain between two household members.

The impact of seasonality on the parasite inbreeding level was assessed by comparing the proportion of related barcodes (IBD ≥0.5) between groups of collection dates within or between transmission seasons. Barcodes sampled during the high transmission seasons of 2014, 2015, and 2016, as well as the low transmission seasons of 2015 and 2016, were grouped into intraseasonal pairs if they were collected during the same season or interseasonal pairs otherwise. This resulted in 5 groups for intraseasonal pairs and 10 groups for interseasonal pairs, with the most distant groups (high 2014 and high 2016) being four seasons apart (*Figure 4A*). As in *Figure 3*, pairs of collection dates close in time exhibited greater similarity than more distant collection dates. This suggests a

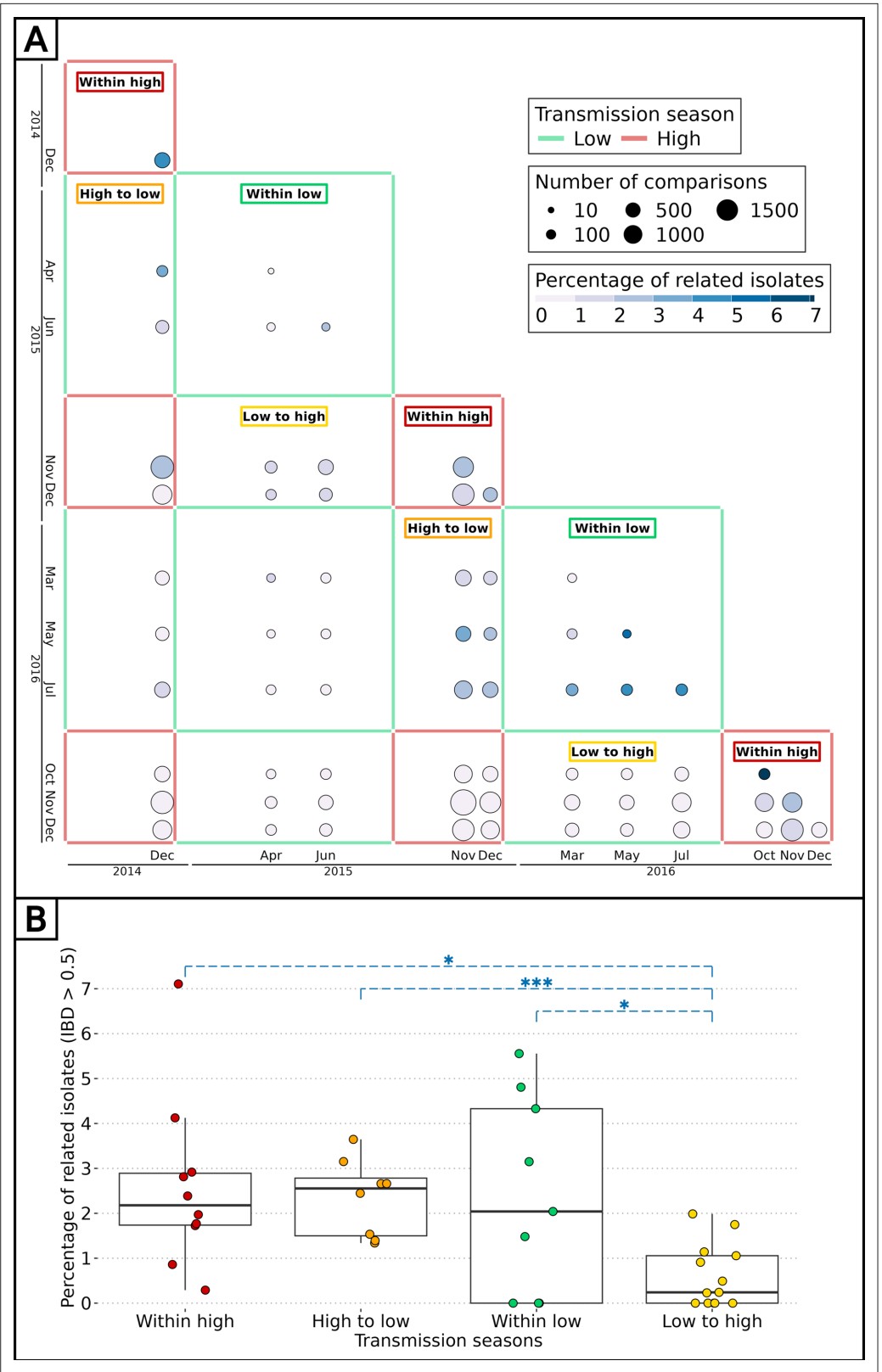

**Figure 4.** Effect of seasonality on parasite inbreeding level. (**A**) Proportion of related barcodes (identity by descent, IBD ≥0.5) between all sample collections from December 2014 to December 2016. Each square represents a set of pairs of time points within the same seasonal pair. There were five successive transmission seasons during the study, hence 15 unique seasonal pairs. Annotations are present for seasonal pairs corresponding to a season

*Figure 4 continued on next page*

*Figure 4 continued*

compared with itself ('within high' and 'within low') and to a season compared with the preceding/succeeding one ('low to high' and 'high to low'). (**B**) Proportion of related barcodes between pairs of sample collections within the same season ('within high' and 'within low') and one season apart when the high transmission season precedes the low transmission season and conversely (respectively 'high to low' and 'low to high'). Box plots display the median, the first and third quartiles, and whiskers extending up to 1.5 times the interquartile range. Genetic similarities were compared between the 'low to high' group and all other groups with Welch's t-tests (*p-value < 0.05, ***p-value < 0.0005).

continuous recombination process among all parasites, rather than the transmission of one or more specific strains. To determine the specific time of the year the average genetic relatedness declines, the proportion of related barcodes was compared between pairs of sample collections from the same season or one season apart (*Figure 4*). Parasites from the 'low to high' group exhibited the lowest average proportion of related barcodes (average relatedness of 0.006) compared with the 'within high' (average relatedness of 0.026, Welch's t-test value = 3.14, p-value < 0.01), 'within low' (average relatedness of 0.024, Welch's t-test value = 2.35, p-value < 0.05) and 'high to low' (average relatedness of 0.024, Welch's t-test value = 4.87, p-value <0.001) groups all displaying a fourfold higher genetic relatedness. Here, most of the parasite differentiation occurs during the transition from the low transmission season to the subsequent high transmission season. This corresponds to the increase in transmission rate at the onset of the high transmission season, with parasite genomes being reshuffled after sexual reproduction in the mosquito. This decrease in relatedness is not observed in the 'high to low' group, demonstrating a reduced transmission in the low transmission season.

## Independence of seasonality and drug resistance markers prevalence

In The Gambia, drug-resistant allele frequencies increased dramatically from the late 1980s until the early 2000s, then plateaued until 2008 (*Nwakanma et al., 2014*). The rise in clinical malaria cases during the high transmission season implies usage of anti-malaria drugs, particularly among children aged between 0 and 5 y receiving antimalarials (sulfadoxine-pyrimethamine and amodiaquine) monthly during Seasonal Malaria Chemoprevention (SMC), from September to November (*World Health Organization, 2012*). The artemisinin-based combination therapy Coartem (artemether-lumefantrine) is the first-line antimalarial used to treat uncomplicated malaria in The Gambia at the time of the study. We investigated whether the prevalence of drug resistance alleles is influenced by malaria seasonality due to differential selective pressures applied between high (more pressure) and low (less pressure) transmission seasons. To determine the prevalence of resistant haplotypes over time, six drug resistance-related haplotypes were obtained by both molecular genotyping (*Supplementary file 8*) and whole genome sequencing (*Supplementary file 9*) in genes *aat1*, *crt*, *dhfr*, *dhps*, *kelch13*, and *mdr1*, leading to a merged total of 438 isolates with drug resistance-related haplotypes. Overall, 89% of haplotypes called from both molecular genotyping and whole genome sequencing were identical (*Figure 5—figure supplement 1*).

As expected, the haplotype Kelch13 C580Y, related to artemisinin resistance, was absent. Overall, the proportion of isolates with resistant alleles was stable over time for AAT1 S258L (0.92, 95% Wilson's Confidence Interval: 0.87–0.95), CRT K76T (0.64, 95 % CI: 0.59–0.70), DHFR S108N (0.93, 95 % CI: 0.90–0.95), and MDR1 N86Y (0.12, 95 % CI: 0.09–0.17) (*Figure 5*). In contrast, the haplotype DHPS A437G appears to decrease in prevalence twice, from the 2015 and 2016 high transmission seasons to the subsequent 2016 and 2017 low transmission seasons. As DHPS A437G haplotype has been associated with resistance to sulfadoxine, its apparent increase in prevalence during high transmission seasons could be resulting from the selective pressure imposed on parasites during SMC. Overall, our data from 2015 to 2017 is very similar to the allele frequency levels from 2008, indicating a potential plateau in the cost-benefit of drug resistance alleles (*Nwakanma et al., 2014*).

## *P. falciparum* chronic infections with persisting genotypes

To distinguish reinfections from 'true' chronic infections with the same parasite genotype, we measured the minimal duration of infection using IBD values between barcodes obtained from the same individual sampled at different time points (which were not utilized for the spatio-temporal analysis of genetic relatedness). The beginning and end of an infection by the same parasite were attributed to

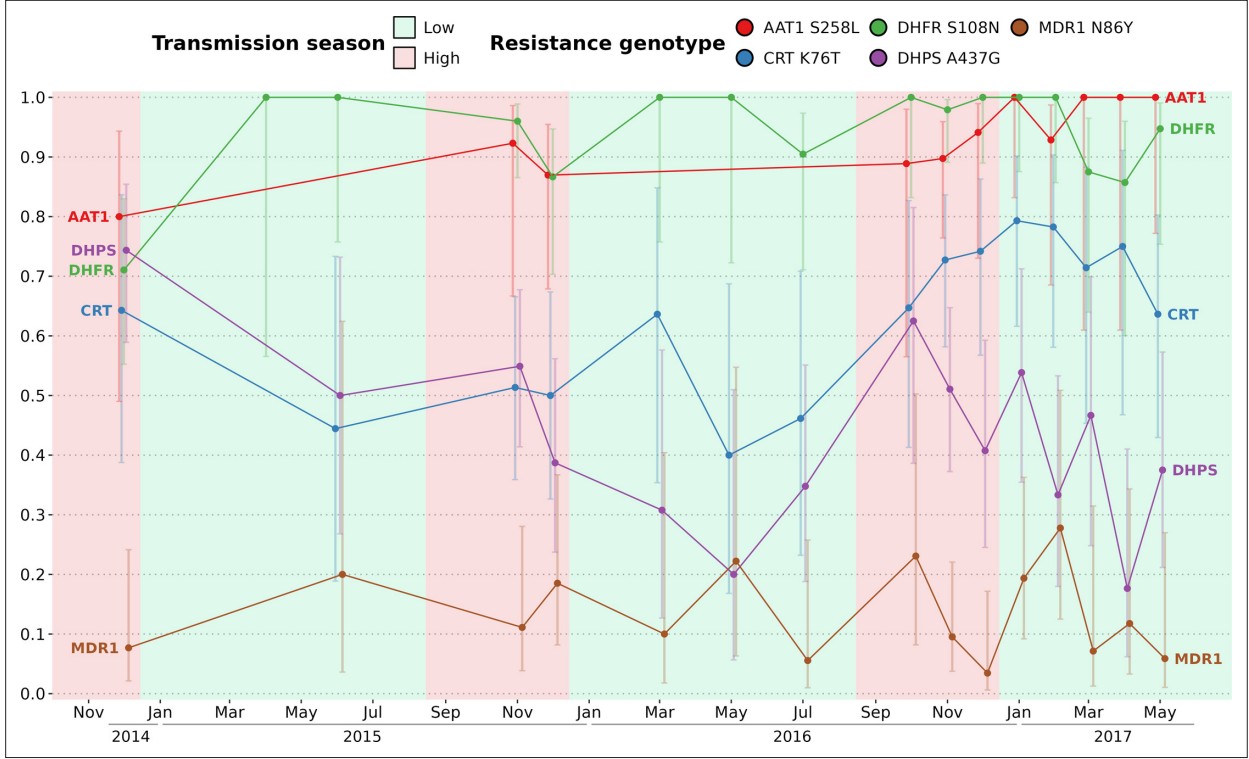

**Figure 5.** Prevalence of five drug resistance-related haplotypes for each time point over the study time period. The 5 variants induce non-synonymous changes in AAT1 S528L, CRT K76T, DHFR S108N, DHPS A437G, and MDR1 N86Y which are known to reduce the susceptibility to multiple antimalarials. The prevalence of the Kelch13 C580Y mutation was null and not shown here. For all markers except DHPS A437G, the proportions of all variants remained stable between the start and end of the study period. Sample sizes range from 5 to 51 (median of 18) with error bars representing the 95% Wilson's confidence intervals.

The online version of this article includes the following figure supplement(s) for figure 5:

**Figure supplement 1.** Agreement between pairs of haplotypes obtained both by molecular genotyping ('barcodes') and from Whole Genome Sequence calling ('genomes').

the two most extreme time points separating two identical barcodes (IBD ≥0.9). Overall, 32 individuals (20 males and 12 females) were chronically infected with the same dominant *P. falciparum* genotype ranging from 1 mo to one and a half years (*Figure 6*, *Supplementary file 10*). Out of the 32 individuals chronically infected, 21 were asymptomatic carriers selected for a monthly sampling between December 2016 and May 2017, raising the chance of observing at least twice the same dominant *P. falciparum* genotype. Gender and age did not significantly influence the duration of infection (*Figure 6—figure supplement 1*). Two individuals (24 and 25) were infected in 2017 with two distinct parasite strains, as shown by monthly barcodes alternating between the two strains (for both individuals, barcodes from Feb and Apr, and from Mar and May, are identical). Interestingly, for individual 24, the two strains were genetically related (IBD ≈ 0.5), likely indicating co-infection of sibling strains from the same brood with a fluctuating relative abundance over time. The change in proportion between the two strains over time might be caused by intra-host competition (e.g. one strain, benefiting from immune evasion, is more abundant than the other) or asynchronous developmental stages (one strain is mostly sequestered when the other is mostly circulating and conversely).

## Discussion

In anticipation of the pre-elimination phase of malaria, our study aimed to develop robust methods for estimating the complexity of infections (COI) and constructing genetic relatedness networks of *P. falciparum* parasites. By tracking 1516 participants over 2.5 y, we found that malaria parasites in this region exhibit high genetic diversity, driven by frequent recombination events rather than dominance by specific strains. Notably, parasites from individuals within the same household were significantly

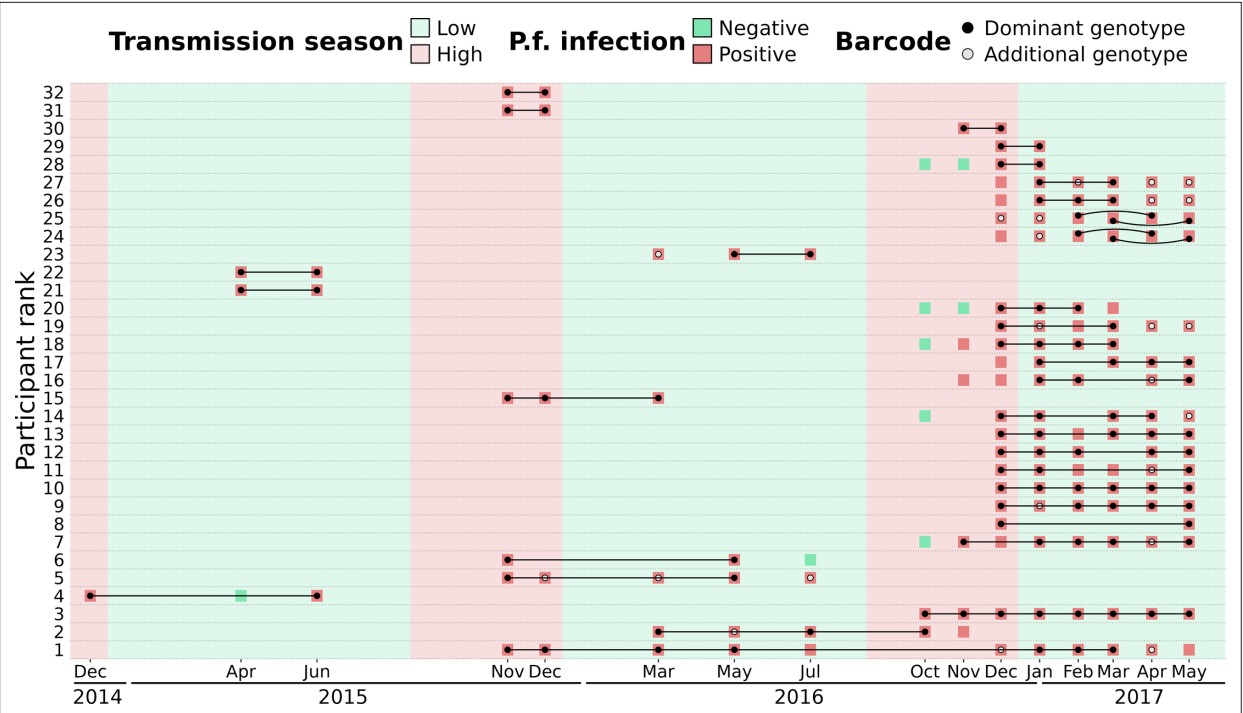

**Figure 6.** Continuous *P. falciparum* infections with the same dominant genotype. A total of 32 individuals were infected with highly related barcodes (identity by descent, IBD ≥0.9) at two or more time points. Individuals are ranked by their duration of continuous infection from the longest to the shortest, with a black line linking identical 'dominant' genotypes (black dots) that are the farthest away from each other. Additional genotypes (gray dots) are different (IBD <0.9) from the dominant genotype but may still be related (IBD ≥0.5). For two individuals (ranked 24th and 25th), two different parasite strains were present concurrently, represented by curved lines. Barcodes and *P. falciparum* infection status are shown up to 90 d prior to or after the inferred continuous infection.

The online version of this article includes the following figure supplement(s) for figure 6:

**Figure supplement 1.** Durations of infection by the same parasite strain, grouped by host age and colored by gender.

more likely to be genetically related, indicating close transmission links within households. Additionally, seasonal changes influenced the genetic relationships among parasites, with populations becoming more diverse during peak transmission periods, reflecting the dynamic nature of malaria transmission in this setting.

Leveraging both barcodes and genomes offers complementary advantages: molecular barcode genotyping provides cost-effective access to genetic information from numerous blood isolates, while whole genomes yield quantitative data on thousands of single nucleotide polymorphisms (SNPs), facilitating comprehensive analyses of parasite diversity, with the added potential to extend analyses to newly discovered haplotypes unidentified at the time of sequencing. Combining both data types validates base calling from molecular genotyping and guides method applicability across the limited positions available in barcodes.

The stable COI over time reflects a constant transmission intensity (*Echeverry et al., 2013*; *Hendry et al., 2021*; *Lee et al., 2021*; *Pacheco et al., 2020*). Such stable transmission in this area of The Gambia, reported already elsewhere, contrasts with the overall decrease of malaria transmission across the country, highlighting the heterogeneity of malaria within The Gambia (*Ahmad et al., 2023*; *Fogang et al., 2024*). Our findings underscore the importance of understanding local malaria dynamics amidst broader regional trends.

With IBD, we evaluated parasite-relatedness between genomes and between barcodes. Although using more SNPs (e.g. more than 200 SNPs) is ideal to reach an accurate relatedness, in silico data suggests that the accuracy is only marginally improved with 96 SNPs or more (*Taylor et al., 2019*). Furthermore, we showed that for IBD values above 0.5, the correlation between genome- and barcode-IBD was very strong. Finally, the qualitative aspect of our approach, with isolates either related (IBD ≥0.5) or unrelated (IBD <0.5) enables to compensate for the lower accuracy of barcode-IBD.

Overall, 26% of unique parasite strains were found in two or more individuals, and around 28% of isolates were polyclonal according to the proportion of heterozygous loci of barcodes. This seemingly contrasts with findings from Thiès, in neighboring Senegal, where parasite haplotype diversity is low, the proportion of shared genetic types between isolates is 35%, with less than 10% of polyclonal infections, suggesting self-mating transmission and low outcrossing levels (*Daniels et al., 2015*; *Redmond et al., 2018*; *Sy et al., 2022*). However, the key difference of our approach was the active detection of asymptomatic infections, as opposed to the large number of studies sequencing parasites collected from clinical cases. Our results argue for active case detection as a necessary step to comprehensively characterize the parasite genetic diversity.

One limitation of genetic epidemiology studies such as this one is the necessity to exclude IBD values obtained between isolates with high level of polyclonality and thus not enough informative loci. One important remaining challenge is to incorporate highly polyclonal infections in the network of genetic relatedness of parasites, while the proportion of complex infections tends to increase with malaria transmission intensity (*Camponovo et al., 2023*). Achieving this goal is difficult as it requires whole genome sequencing followed by a deconvolution tool such as DEploid, with the caveat that the different strains are not too related or too rare (*Nkhoma et al., 2020*; *Zhu et al., 2018*). A second limitation to our study is that we cannot exclude that imported malaria cases might have an effect on our measured inbreeding levels. However, previous findings tend to indicate a limited effect of imported cases on maintaining transmission (*Daniels et al., 2015*; *Fola et al., 2023*; *Gwarinda et al., 2021*; *Sy et al., 2022*). Future studies will require recruiting a larger number of participants with more frequent samplings.

A follow-up study carried out in rural villages of eastern Gambia between 2012 and 2016 reported that roughly half of asymptomatic infected individuals at the end of the wet season were still infected at the end of the dry season (*Ahmad et al., 2023*). We previously reported a similar rate (*Collins et al., 2022*). We showed that parasites collected during the dry season share significantly more genetic similarity with those from the previous wet season than those from the next wet season. Average relatedness quickly declined with the temporal distance between samplings (10-fold lower after 1 y), probably due to active genetic recombinations in the population leading to fewer fragments in IBD over time. A study conducted in Colombia (with a 10-fold lower malaria prevalence than in The Gambia) showed that a 10-fold decrease in genetic similarity was reached in 9 y while we observed the same decrease after just 1 y in our study (*Echeverry et al., 2013*; *Taylor et al., 2020*).

Most infections were sub-microscopic, which is typically observed when prevalence falls below 20% (*Fogang et al., 2024*; *Okell et al., 2012*; *Slater et al., 2019*). These low-density infections that may persist for months are typically asymptomatic unless the host immune system is compromised (*Ashley and White, 2014*). In this study, the infection with the longest duration was in an asymptomatic individual aged between 5 and 9 y in which it persisted as the same parasite strain for one and a half year. School-age children contribute the most to the malaria reservoir through their higher carriage of asymptomatic infections and their ability to effectively infect mosquitoes (*Ahmad et al., 2023*; *Andolina et al., 2021*; *Bylicka-Szczepanowska and Korzeniewski, 2022*; *Mueller et al., 2012*).

In the late 20th century, when the malaria burden in The Gambia was much higher, clinical cases from the same household were more likely to be caused by the same mosquito bite (*Conway and McBride, 1991*). More recently, Ngwa et al. showed that parasite genetic similarity during the 2013 transmission season was inversely correlated with each spatial and temporal distances in The Gambia (*Amambua-Ngwa et al., 2019*). We also found that parasites sampled in neighbouring households are more genetically similar but only when sampled less than 3 mo apart. Similarly, two other studies observed parasite strains with varying levels of spatio-temporal propagation in Thiès, Senegal, suggesting that some parasites are more actively transmitted than others (*Lee et al., 2021*; *Sy et al., 2022*). These results add evidence that anti-malarial strategies should target all members of a household with an infected individual. Such 'reactive' strategies include treatment of a malaria case and all its household members (without testing) or testing all household members and treating if necessary. The impact of reactive strategies on malaria transmission is at best very limited (*Newby et al., 2023*; *Okebe et al., 2021*; *Steinhardt et al., 2024*; *Stresman et al., 2020*). Based on the sheer size of the *P. falciparum* asymptomatic reservoir, with parasitemias typically below microscopy or RDT detection level (*Fogang et al., 2024*), it is not surprising that targeting clinical cases and their immediate families is not sufficient to break the transmission. For a malaria elimination campaign, based on our

insights and previous research (*Soremekun et al., 2024*) on asymptomatic infections in The Gambia at the community-level, we argue for mass detection with a highly sensitive method such as qPCR, followed by treatment of *P. falciparum* positive cases and all their household members.

## Materials and methods

### Study design and participants

Starting in December 2014, we recruited all residents from two villages (Madina Samako and Njayel, identified, respectively, with the letters 'K' and 'J'), with two additional villages (Sendebu and Karandaba, identified respectively with the letters 'P' and 'N') recruited from July 2016, all four villages being in the Upper River Region in The Gambia within 5 km of each other (*Supplementary file 1*). Active case detection was conducted a total of 11 times over the 2 y period, with each sampling session occurring within a 10-d window (*Supplementary file 2*). Symptomatic cases that occurred between September and December 2016 were also sampled. More information about the recruited participants can be found in a previous study (*Fogang et al., 2024*). In December 2016, 74 asymptomatic *P. falciparum* carriers were sampled monthly for up to 6 mo until May 2017, with 42 of them being part of a cohort previously reported (*Collins et al., 2022*).

The study protocol was reviewed and approved by the Gambia Government/MRC Joint Ethics Committee (SCC 1476, SCC 1318, L2015.50) and by the London School of Hygiene & Tropical Medicine Ethics Committee (Ref. 10982). The field studies were also approved by local administrative representatives, the village chiefs. Written informed consent was obtained from participants over 18 y old and from parents/guardians for participants under 18 y. Written assent was obtained from all individuals aged 12–17 y. Approval under the Nagoya Protocol was granted by the National Focal Point of The Gambia.

### Sampling and molecular detection of parasites

Details on the *P. falciparum* detection is provided in previous works (*Collins et al., 2022*; *Fogang et al., 2024*). Briefly, fingerprick blood samples were tested for *P. falciparum* by varATS qPCR. From July 2016 onwards, individuals testing positive for *P. falciparum* were invited to provide an additional 5–8 mL venous blood sample that was leucodepleted with cellulose-based columns (MN2100ff) and frozen immediately, as described by MalariaGEN (https://www.malariagen.net/article/online-protocol-leucocyte-depletion-using-mn2100ff-cellulose-columns/). DNA was extracted with the QIAgen Miniprep kit following the manufacturer procedure.

### Genotyping and genome sequencing

A total of 522 *P. falciparum* DNA positive samples, 307 from fingerprick and 215 from venous blood, were processed for genotyping (442 samples) and whole genome sequencing when sufficient parasite DNA was available (331 samples of which 251 were genotyped) as part of the SpotMalaria consortium (*Figure 1B*). Genotyping was performed by mass-spectrometry-based platform from Agena MassArray system. The output consisted of 101 bi-allelic SNPs located on the 14 chromosomes and concatenated into a 'molecular barcode' (*Supplementary file 3*; *Supplementary file 4*; *Jacob et al., 2021*), plus six markers of resistance to antimalarials: AAT1 S528L (associated with chloroquine resistance), CRT K76T (associated with chloroquine resistance), DHFR S108N (associated with pyrimethamine resistance), DHPS A437G (associated with sulfadoxine resistance), Kelch13 C580Y (associated with artemisinin resistance), and MDR1 N86Y (associated with chloroquine, amodiaquine, lumefantrine and mefloquine resistance). The 101 SNPs had been picked for their variable allele frequencies within the *P. falciparum* population (*Jacob et al., 2021*). Whole genome sequencing (Illumina) was performed after a Selective Whole Genome Amplification step (*Oyola et al., 2016*). Paired-end DNA sequence reads (150 bp) were aligned to 3D7 reference genome version 3. Variants were called by a script from the MalariaGen consortium using GATK HaplotypeCaller (*MalariaGEN et al., 2021*; *McKenna et al., 2010*). Two drug resistance markers, AAT1 S528L and Kelch13 C580Y were only available in sequencing data and absent from genotyping data. If distinct calls of drug resistance markers were obtained between genotyping and whole genome sequencing data, only the call of whole genome sequencing was considered for the rest of the analysis.

## Parasite relatedness

To accurately assess the parasite genetic similarity between different sampled infections, we estimated pairwise mean posterior probabilities of IBD between genomes or barcodes using hmmIBD, a hidden Markov model-based software relying on meiotic recombination events given a recombination rate of *Plasmodium falciparum* of 13.5 kb/cM (*Miles et al., 2016*; *Schaffner et al., 2018*). To estimate IBD, the model requires pairs of homozygous sites between two samples that are referred to as 'comparable sites' in this manuscript. These comparable sites are used by hmmIBD to yield the probability of two samples to be in IBD, which is equivalent to the expected shared fraction of their genomes. With an IBD of more than 0.9, two samples are considered identical, hence describing the same parasite genotype.

When samples are highly polyclonal, their set of homozygous sites tend to be made out of mostly alleles nearly fixed in the population, with a resulting IBD artificially high. Setting a minimal number of comparable sites for which at least one sample possess the minor allele will reduce this bias; these sites will be referred to as 'informative sites' in this manuscript.

## Multi-locus genotype barcode data analysis pipeline

We developed a pipeline in R (version 4.3.3, *R Development Core Team, 2024*) combining Whole Genome Sequencing (WGS) and molecular genotyping data to produce a genetic relatedness network of parasites.

To analyze WGS data formatted in a VCF format, we developed the following pipeline to (*Figure 1—figure supplement 3*):

1. Filter out genomic loci for which the population minor allele frequency is inferior to 0.01, with more than 2 alleles identified in the population or that are located outside of the core genome (*Miles et al., 2016*). Out of 1,042,186 initial SNPs, 27,577 (minimal QUAL of 132) were retained.
2. Remove genomes comprising less than 3000 SNPs covered by at least 5 reads.
3. Estimate the proportion of polyclonal samples using the $F_{ws}$ metric and heterozygous loci.
4. Format SNPs into a matrix as required by hmmIBD. The matrix was formatted as follows: 0 and 1 were used for the respective allele of each SNP and –1 for mixed and unknown positions (which were then ignored during the IBD estimation). SNP calls were considered mixed if the within-sample Minor Allele Frequency (MAF) was greater than 0.2. The MAF of 0.2 was chosen according to the good agreement between molecular barcodes and genomic barcodes (high number of heterozygous call matches and low number of heterozygous-homozygous call mismatches) (*Figure 1—figure supplement 6*).
5. Use the paired IBD values obtained from hmmIBD and build a network of genome relatedness. IBD values were considered unknown between pairs of genomes having less than 100 informative sites (all our pairs had at least 1000 comparable sites).

The second part of the pipeline imputes missing SNPs in the molecular barcode from the WGS data to build a 'consensus barcode' (*Figure 1—figure supplement 3*):

1. Build a 'molecular barcode' out of the initial 101 genotyped SNPs and a 'genomic barcode' out of the same 101 SNPs called from high-quality genomes.
2. Remove 12 loci that are absent from all high-quality genomes.
3. Estimate the within-sample MAF to use as a cutoff to consider a locus homozygous when building barcodes out of genomic data. Genomic barcodes are built using different cutoffs of MAF below which a locus is considered homozygous and aligned against molecular barcodes from the same isolates. The cutoff of within-sample MAF of 0.2 showed a high number of heterozygous call matches and low number of heterozygous-homozygous call mismatches, meaning that molecular barcodes and genomic barcodes were in good agreement (*Figure 1—figure supplement 6*). This cutoff was retained to build all genomic barcodes. Each pair of molecular and genomic barcodes obtained from the same isolates were aligned. For isolates sampled after May 2016, molecular barcodes are most of the time not matching genomic barcodes for 21 loci (*Figure 1—figure supplement 7*). These 21 loci were clustered in 1 of the 4 multiplexes used for molecular genotyping that probably failed to give the proper base calling. As a result, these 21 loci were considered unknown for all the molecular barcodes obtained after May 2016.
4. Using the alignment between molecular and genomic barcodes from the same isolates, replace unknown and mismatched SNP of the molecular barcodes by the SNP of genomic barcodes.

These improved molecular barcodes are referred to as 'consensus barcodes' (***Supplementary files 3 and 4***).

5. Remove consensus barcodes with fewer than 30 SNPs.
6. Estimate the proportion of polyclonal samples using heterozygous loci (described in the following section).
7. Format consensus barcodes into a matrix as required by hmmIBD. The matrix was formatted like with genome data.
8. Use the paired IBD values obtained from hmmIBD and build a network of barcode relatedness. IBD values were considered unknown between pairs of barcodes having less than 30 comparable sites and 10 informative sites.

## Complexity of infections

The clonality of each isolate was estimated from whole genome sequenced samples by the $F_{ws}$ metric based on allelic frequencies from genomic data (***Manske et al., 2012***). Additionally, the complexity of infections was estimated by the proportion of heterozygous loci (polyclonal if the proportion is above 0.5% of available sites) in both consensus barcodes and genomes using a within-sample MAF of 0.2.

## Genetic relatedness between groups of multi-locus genotype barcodes

All 'consensus barcodes' (hereafter referred to as 'barcodes') available after December 2016 were excluded from the genetic relatedness analysis as they were obtained exclusively from the 74 individuals selected for their asymptomatic carriage of *P. falciparum* (***Collins et al., 2022***).

IBD was used to classify barcode ('barcode-IBD') pairs as related (IBD ≥0.5) or unrelated (IBD <0.5). This classification was compared to the one resulting from IBD calculated between pairs of genomes ('genome-IBD,' considered as gold-standard) using Cohen's kappa (***McHugh, 2012***). For pairs of isolates with a different classification between barcode-IBD and genome-IBD, only the genome-IBD classification was retained (57/364 falsely related pairs set as unrelated and 58/18626 falsely unrelated pairs set as related). Additionally, the genome-IBD was used to classify barcodes as related or unrelated when the barcode-IBD was not available (710 pairs).

Barcodes from 11 time points (December 2014 to December 2016) were grouped by their collection date, sampling location (households or villages) or collection date split by household groups (within the same household, two households of the same village or two households of different villages). Within the same group, the genetic relatedness was estimated by the proportion of related barcodes (IBD ≥0.5) over all possible pairs of barcodes. When comparing different groups, the genetic relatedness was estimated by the proportion of related barcodes over all pairs of barcodes of each group, excluding pairs involving barcodes from the same individual. We used a stringent criteria of filtering out pairs of collection dates split by household groups with less than five comparisons (25/192 removed pairs). All pairs of villages (10 pairs) and all pairs of dates (66 pairs) contained at least 10 comparisons.

## Acknowledgements

The authors would like to thank the staff of MalariaGEN, Wellcome Sanger Institute Sample Management, Genotyping, Sequencing, and Informatics teams for their contribution. We thank Michael Fontaine, Franck Prugnolle, and Virginie Rougeron for insightful comments on data analysis. This work was funded by grants from the Netherlands Organization for Scientific Research (Vidi fellowship NWO 016.158.306) and the Bill & Melinda Gates Foundation (INDIE OPP1173572), the joint MRC/LSHTM fellowship, CNRS Transversales, the French National Research Agency (ANR 18-CE15-0009-01), the Fondation pour la Recherche Médicale (EQU202303016290).

## Additional information

### Funding

| Funder | Grant reference number | Author |
|---|---|---|
| Netherlands Organisation for Scientific Research | NWO 016.158.306 | Teun Bousema |
| Bill and Melinda Gates Foundation | INDIE OPP1173572 | Teun Bousema |
| Medical Research Council | MRC/LSHTM fellowship | Antoine Claessens |
| London School of Hygiene and Tropical Medicine | MRC/LSHTM fellowship | Antoine Claessens |
| Centre National de la Recherche Scientifique | Transversales | Marc-Antoine Guery |
| Agence Nationale de la Recherche | ANR 18-CE15-0009-01 | Antoine Claessens |
| Fondation pour la Recherche Médicale | EQU202303016290 | Antoine Claessens |

The funders had no role in study design, data collection and interpretation, or the decision to submit the work for publication.

### Author contributions

Marc-Antoine Guery, Conceptualization, Data curation, Software, Formal analysis, Investigation, Visualization, Writing – original draft, Writing – review and editing; Sukai Ceesay, Sainabou Drammeh, Fatou K Jaiteh, Methodology; Umberto D'Alessandro, Resources, Supervision, Writing – review and editing; Teun Bousema, Conceptualization, Resources, Funding acquisition, Writing – review and editing; David J Conway, Conceptualization, Formal analysis, Supervision, Writing – review and editing; Antoine Claessens, Conceptualization, Resources, Formal analysis, Supervision, Funding acquisition, Investigation, Writing – original draft, Project administration

### Author ORCIDs

Umberto D'Alessandro ⓘ https://orcid.org/0000-0001-6341-5009
Teun Bousema ⓘ https://orcid.org/0000-0003-2666-094X
Antoine Claessens ⓘ https://orcid.org/0000-0002-4277-0914

### Ethics

The study protocol was reviewed and approved by the Gambia Government/MRC Joint Ethics Committee (SCC 1476, SCC 1318, L2015.50) and by the London School of Hygiene & Tropical Medicine Ethics Committee (Ref. 10982). The field studies were also approved by local administrative representatives, the village chiefs. Written informed consent was obtained from participants over 18 years old and from parents/guardians for participants under 18 years. Written assent was obtained from all individuals aged 12-17 years. Approval under the Nagoya Protocol was granted by the National Focal Point of The Gambia.

Reviewer #2 (Public review): https://doi.org/10.7554/eLife.103047.3.sa1
Reviewer #3 (Public review): https://doi.org/10.7554/eLife.103047.3.sa2
Author response https://doi.org/10.7554/eLife.103047.3.sa3

## Additional files

### Supplementary files

Supplementary file 1. Gender, age, and households of participants.
Supplementary file 2. Sampling details, *falciparum*-malaria test results and treatment data.
Supplementary file 3. Genomic location of single nucleotide polymorphisms (SNPs) and their order

within barcodes.

Supplementary file 4. Raw data and summary statistics from molecular barcode genotyping. Barcode_ID is the sample identifier used in this study.

Supplementary file 5. Genome sequencing accession identifiers and statistics. Genome_ID is the sample identifier used in this study. VCF_ID is the corresponding identifier in MalariaGEN Pf6 release. ERR_ID is the corresponding identifier in the European Nucleotide Archive (ENA).

Supplementary file 6. Complexity of infection metrics from barcode data.

Supplementary file 7. Complexity of infection metrics from genome data.

Supplementary file 8. Drug resistance markers genotyped from barcode data.

Supplementary file 9. Drug resistance markers genotyped from genome data.

Supplementary file 10. Duration of continuous infection estimated from barcode relatedness.

MDAR checklist

### Data availability

The barcode data analysis pipeline can be found at https://github.com/marcguery/malaria-barcodes-genomes-gambia (copy archived at *Guery, 2025*). This publication uses data from the MalariaGEN *Plasmodium falciparum* Community Project as described in 'An open dataset of Plasmodium falciparum genome variation in 7,000 worldwide samples (*MalariaGEN et al., 2021*). Molecular barcode genotyping data are available in *Supplementary files 3 and 4*. Whole genome sequencing data are accessible in the European Nucleotide Archive (https://www.ebi.ac.uk/ena/browser/view) and MalariaGEN Pf6 release (https://www.malariagen.net/resource/26/) using the accession numbers of *Supplementary file 5*.

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
