## [Editor Report · eLife Assessment]

This article presents a **valuable** genetic spatio-temporal analysis of malaria-infected individuals from four villages in a highly seasonal transmission setting in The Gambia, covering the period between December 2014 and May 2017. Evidence generated by the study's laboratory and data processing approaches is **solid** and helps to advance the understanding of malaria in The Gambia, particularly due to its longitudinal design and the inclusion of asymptomatic cases.

---

## [Referee Report · Reviewer #2 (Public review)]

Summary:

Malaria transmission in the Gambia is highly seasonal, whereby periods of intense transmission at the beginning of the rainy season are interspersed by long periods of low to no transmission. This raises several questions about how this transmission pattern impacts the spatiotemporal distribution of circulating parasite strains, how parasites persist during the dry season, and how asymptomatic infections contribute to maintaining transmission during the low/no transmission season.

Combining a molecular barcode genotyping using 101 bi-allelic SNPs and SNPs from Whole Genome Sequence (WGS) in a "consensus barcode", the authors aimed at measuring the relatedness between parasites at different spatial (i.e., individual, household, village, and region) and temporal (i.e., high, low, and the corresponding the transitions) levels by assessing the fraction of the genome having a common ancestry (i.e. Identity-by-Descent (IBD)).

By measuring the Complexity of Infection (COI) and parasite relatedness by IBD the authors show that a large fraction of infections is polygenomic and stable over time, resulting in a high recombinational diversity. Moreover, they show that transmission intensity increases during the transition from the dry to wet seasons. However, they find that there is a higher probability of finding similar genotypes within the same household, but this similarity rapidly disappears over time and is not observed between different villages. If there is no drug selection during the dry season, and if resistance results in a fitness cost, alleles associated with drug resistance may change in frequency. The authors looked at the frequencies of six drug-resistance haplotypes (aat1, crt, dhfr, dhps, kelch13, and mdr1), and found no evidence of changes in allele frequencies associated with seasonality. They also find chronic infections lasting from one month to one and a half years with no dependence on age or gender.

This work makes use of genomic information and IBD analytic tools to show parasite relatedness from asymptomatic infections at different spatial and temporal scales, thus providing a better understanding of the transmission dynamics of malaria in highly seasonal environments.

Strength:

The authors use a combination of high-quality barcodes (425 barcodes representing 101 bi-allelic SNPs) and 199 high-quality genome sequences to infer the fraction of the genome with shared Identity by Descent (IBD) (i.e. a metric of recombination rate) over several time points covering two years. The barcode and whole genome sequence combination allows full use of a large dataset, to confidently infer the relatedness of parasite isolates at various spatiotemporal scales and show the advantage of using genomic information for understanding malaria transmission dynamics.

The authors aimed to establish how seasonal transmission cycles shape the spatiotemporal parasite population structure using metrics such as parasite genetic diversity, genetic relatedness, and frequency of drug resistance alleles, as well as the contribution of asymptomatic chronic carriers to sustained transmission. The results support their conclusions.

Using a combination of molecular barcodes and available whole genome sequence datasets opens new opportunities to understand malaria transmission dynamics in different transmission settings. This allows for data analysis at different spatiotemporal granularities, having a practical utility for identifying malaria control targets and acquiring metrics to evaluate malaria control programs. The development of molecular barcodes using similar SNPs by different malaria control programs would be of great utility to compare and understand malaria transmission dynamics in different settings worldwide.

---

## [Referee Report · Reviewer #3 (Public review)]

This study aimed to examine the impact of seasonality on the population genetics of malaria parasites. To achieve this, the researchers conducted a longitudinal study in a region with seasonal malaria transmission. Over a 2.5-year period, blood samples were collected from 1,516 participants across four villages in the Upper River Region of The Gambia. These samples were tested for malaria parasite infection, and the parasites from positive samples were genotyped using a genetic barcode and/or whole genome sequencing. Genetic relatedness analysis was then performed to explore the findings

The study identified three key findings:

(1) The malaria parasite population undergoes continuous recombination, with no single genotype predominating, in contrast to viral populations;

(2) Parasite relatedness is influenced by both spatial and temporal factors; and

(3) The lowest genetic relatedness among parasites occurs during the transition from the low to high transmission seasons, which the authors linked to increased recombination during sexual reproduction in mosquitoes.

The results section is well-structured, and the figures are clear and self-explanatory. The methods are adequately described, providing a solid foundation for the findings. While there are no unexpected results, it is reassuring to see the anticipated outcomes supported by actual data. The conclusions are generally well-supported and the recommendation to target asymptomatic infections is logical and relevant.

---

## [Author Response]

The following is the authors’ response to the original reviews

**Reviewer #1 (Public review):**
Summary:The manuscript titled "Household clustering and seasonal genetic variation of *Plasmodium falciparum* at the community-level in The Gambia" presents a valuable genetic spatio-temporal analysis of malaria-infected individuals from four villages in The Gambia, covering the period between December 2014 and May 2017. The majority of samples were analyzed using a SNP barcode with the Spotmalaria panel, with a subset validated through WGS. Identity-by-descent (IBD) was calculated as a measure of genetic relatedness and spatio-temporal patterns of the proportion of highly related infections were investigated. Related clusters were detected at the household level, but only within a short time period.Strengths:This study offers a valuable dataset, particularly due to its longitudinal design and the inclusion of asymptomatic cases. The laboratory analysis using the Spotmalaria platform combined and supplemented with WGS is solid, and the authors show a linear correlation between the IBD values determined with both methods, although other studies have reported that at least 200 SNPs are required for IBD analysis. Data-analysis pipelines were created for (1) variant filtering for WGS and subsequent IBD analysis, and (2) creating a consensus barcode from the spot malaria panel and WGS data and subsequent SNP filtering and IBD analysis.Weaknesses:Further refining the data could enhance its impact on both the scientific community and malaria control efforts in The Gambia.(1) The manuscript would benefit from improved clarity and better explanation of results to help readers follow more easily. Despite familiarity with genotyping, WGS, and IBD analysis, I found myself needing to reread sections. While the figures are generally clear and well-presented, the text could be more digestible. The aims and objectives need clearer articulation, especially regarding the rationale for using both SNP barcode and WGS (is it to validate the approach with the barcode, or is it to have less missing data?). In several analyses, the purpose is not immediately obvious and could be clarified.

The text of the manuscript has now been thoroughly revised. But please let us know if a specific section remains unclear.

(2) Some key results are only mentioned briefly in the text without corresponding figures or tables in the main manuscript, referring only to supplementary figures, which are usually meant for additional detail, but not main results. For example, data on drug resistance markers should be included in a table or figure in the main manuscript.

We agree with the reviewer suggesting to move the prevalence of drug resistance markers from supplementary figures (previously Figure S8) to the main manuscript (now Figure 5). If other Figure/Table should be moved to the main manuscript please let us know.

(3) The study uses samples from 2 different studies. While these are conducted in the same villages, their study design is not the same, which should be addressed in the interpretation and discussion of the results. Between Dec 2014 and Sept 2016, sampling was conducted only in 2 villages and at less frequent intervals than between Oct 2016 to May 2017. The authors should assess how this might have impacted their temporal analysis and conclusions drawn. In addition, it should be clarified why and for exactly in which analysis the samples from Dec 2016 - May 2017 were excluded as this is a large proportion of your samples.

We have clarified which set of samples was used in our Results (Lines 293-295, 316-319). While two villages were recruited halfway through the study, two villages (J and K, Figure 1C) consistently provided data for each transmission season. Importantly, our temporal analysis accounts for these differences by grouping paired barcodes based on their respective locations (Figure 3B). Despite variations in sampling frequency, we still observe a clear overall decline in relatedness between the ‘0-2 months’ and ‘2-5 months’ groups, both of which include barcodes from all four villages.

(4) Based on which criteria were samples selected for WGS? Did the spatiotemporal spread of the WGS samples match the rest of the genotyped samples? I.e. were random samples selected from all times and places, or was it samples from specific times/places selected for WGS?

All *P. falciparum* positive samples were sent for genotyping and whole genome sequencing, ensuring no selection bias. However, only samples with sufficient parasite DNA were successfully sequenced. We have updated the text (Line 129-130) and added a supplementary figure (Figure S4) to show the sample collection broken down by type of data (barcode or genome). High quality genomes are distributed across all time points.

(5) The manuscript would benefit from additional detail in the methods section.

Please see our response in the section “Recommendation for the authors”.

(6) Since the authors only do the genotype replacement and build consensus barcode for 199 samples, there is a bias between the samples with consensus barcode and those with only the genotyping barcode. How did this impact the analysis?

While we acknowledge the potential for bias between samples with a consensus barcode (based on WGS) and those with genotyping-only barcodes, its impact is minimal. WGS does indeed produce a more accurate barcode compared to SNP genotyping, but any errors in the genotyping barcodes were mitigated by excluding loci that systematically mismatched with WGS data (see Figure S3). Additionally, the use of WGS improved the accuracy of 51 % (216/425) of barcodes, which strengthens the overall quality and validity of our analysis.

(7) The linear correlation between IBD-values of barcode vs genome is clear. However, since you do not use absolute values of IBD, but a classification of related (>=0.5 IBD) vs. unrelated (<0.5), it would be good to assess the agreement of this classification between the 2 barcodes. In Figure S6 there seem to be quite some samples that would be classified as unrelated by the consensus barcode, while they have IBD>0.5 in the Genome-IBD; in other words, the barcode seems to be underestimating relatedness.a. How sensitive is this correlation to the nr of SNPs in the barcode?

We measured the agreement between the two classifications using specificity (0.997), sensitivity (0.841) and precision (0.843) described in the legend of Figure S8. To further demonstrate the good agreement between the two methods, we calculated a Cohen’s kappa value of 0.839 (Lines 226, 290), indicative of a strong agreement (McHugh 2012). As expected, the correlation between IBD values obtained by both methods improves (higher Cohen’s kappa and R^2^) as the cutoff for the minimal number of comparable and informative loci per barcode pair is raised (data not shown).

(8) With the sole focus on IBD, a measure of genetic relatedness, some of the conclusions from the results are speculative.a. Why not include other measures such as genetic diversity, which relates to allele frequency analysis at the population level (using, for example, nucleotide diversity)? IBD and the proportion of highly related pairs are not a measure of genetic diversity. Please revise the manuscript and figures accordingly.

We agree with the fact that IBD is not a direct measure of genetic diversity, even though both are related (Camponovo et al., 2023). More precisely, IBD is a measure of the level of inbreeding in the population (Taylor et al., 2019). We have updated our manuscript by replacing “genetic diversity” with “genetic relatedness” or “inbreeding/outcrossing” when appropriate. Nucleotide diversity would be relevant if we wanted to compare different settings, e.g. Africa vs Asia, however this is not the case here.

b. Additionally, define what you mean by "recombinatorial genetic diversity" and explain how it relates to IBD and individual-level relatedness.

We considered the term ‘recombinatorial genetic diversity’ to be equivalent to the level of inbreeding in the population. Because this expression is rather uncommon, we decided to drop it from our manuscript and replace it with “inbreeding/outcrossing”.

c. Recombination is one potential factor contributing to the loss of relatedness over time. There are several other factors that could contribute, such as mobility/gene flow, or study-specific limitations such as low numbers of samples in the low transmission season and many months apart from the high transmission samples.

Indeed, the loss of relatedness could be attributed not only to the recombination of local cases but also to new parasites introduced by imported malaria cases. As we stated in our manuscript, previous studies have shown a limited effect of imported cases on maintaining transmission (Lines 72-74). Nevertheless, we cannot definitely exclude that imported cases have an effect on inbreeding levels, since we do not have access to genetic data of surrounding parasites at the time of the study. We updated the discussion accordingly (Lines 497-501).

d. By including other measures such as linkage disequilibrium you could further support the statements related to recombination driving the loss of relatedness.

This commendable suggestion is actually part of an ongoing project focusing on the sharing of IBD fragments and how it correlates with linkage disequilibrium. However, we believe that this analysis would not fit in the scope of our manuscript which is really about spatio-temporal effects on parasite relatedness at a local scale.

(9) While the authors conclude there is no seasonal pattern in the drug-resistant markers, one can observe a big fluctuation in the dhps haplotypes, which go down from 75% to 20% and then up and down again later. The authors should investigate this in more detail, as dhps is related to SP resistance, which could be important for seasonal malaria chemoprofylaxis, especially since the mutations in dhfr seem near-fixed in the population, indicating high levels of SP resistance at some of the time points.

As the reviewer noted, the DHPS A437G haplotype appears to decrease in prevalence twice throughout our study: from the 2015 and 2016 high transmission seasons to the subsequent 2016 and 2017 low transmission seasons. Seasonal Malaria Chemoprophylaxis (SMC) was carried out in the area through the delivery of sulfadoxine–pyrimethamine plus amodiaquine to children 5 years old and younger during high transmission seasons. As DHPS A437G haplotype has been associated with resistance to sulfadoxine, its apparent increase in prevalence during high transmission seasons could be resulting from the selective pressure imposed on parasites. After SMC, the decrease in prevalence observed during low transmission seasons could be caused by a fitness cost of the mutation favouring wild-type parasites over resistant ones. We updated our manuscript to reflect this relevant observation (Lines 400-405).

(10) I recommend that raw data from genotyping and WGS should be deposited in a public repository.

Genotyping data is available in the supplementary table 4 (Table S4). Whole genome sequencing is accessible in a European Nucleotide Archive public repository with the identifiers provided in supplementary table 5 (Table S5). We added references to these tables in the manuscript (Lines 249-250).

**Reviewer #2 (Public review):**
Summary:Malaria transmission in the Gambia is highly seasonal, whereby periods of intense transmission at the beginning of the rainy season are interspersed by long periods of low to no transmission. This raises several questions about how this transmission pattern impacts the spatiotemporal distribution of circulating parasite strains. Knowledge of these dynamics may allow the identification of key units for targeted control strategies, the evaluation of the effect of selection/drift on parasite phenotypes (e.g., the emergence or loss of drug resistance genotypes), and analyze, through the parasites' genetic nature, the duration of chronic infections persisting during the dry season. Using a combination of barcodes and whole genome analysis, the authors try to answer these questions by making clever use of the different recombination rates, as measured through the proportion of genomes with identity-by-descent (IBD), to investigate the spatiotemporal relatedness of parasite strains at different spatial (i.e., individual, household, village, and region) and temporal (i.e., high, low, and the corresponding the transitions) levels. The authors show that a large fraction of infections are polygenomic and stable over time, resulting in high recombinational diversity (Figure 2). Since the number of recombination events is expected to increase with time or with the number of mosquito bites, IBD allows them to investigate the connectivity between spatial levels and to measure the fraction of effective recombinational events over time. The authors demonstrate the epidemiological connectivity between villages by showing the presence of related genotypes, a higher probability of finding similar genotypes within the same household, and how parasite-relatedness gradually disappears over time (Figure 3). Moreover, they show that transmission intensity increases during the transition from dry to wet seasons (Figure 4). If there is no drug selection during the dry season and if resistance incurs a fitness cost it is possible that alleles associated with drug resistance may change in frequency. The authors looked at the frequencies of six drug-resistance haplotypes (aat1, crt, dhfr, dhps, kelch13, and mdr1), and found no evidence of changes in allele frequencies associated with seasonality. They also find chronic infections lasting from one month to one and a half years with no dependence on age or gender.The use of genomic information and IBD analytic tools provides the Control Program with important metrics for malaria control policies, for example, identifying target populations for malaria control and evaluation of malaria control programs.Strength:The authors use a combination of high-quality barcodes (425 barcodes representing 101 bi-allelic SNPs) and 199 high-quality genome sequences to infer the fraction of the genome with shared Identity by Descent (IBD) (i.e. a metric of recombination rate) over several time points covering two years. The barcode and whole genome sequence combination allows full use of a large dataset, and to confidently infer the relatedness of parasite isolates at various spatiotemporal scales.
**Reviewer #3 (Public review):**
SummaryThis study aimed to investigate the impact of seasonality on the malaria parasite population genetic. To achieve this, the researchers conducted a longitudinal study in a region characterized by seasonal malaria transmission. Over a 2.5-year period, blood samples were collected from 1,516 participants residing in four villages in the Upper River Region of The Gambia and tested the samples for malaria parasite positivity. The parasites from the positive samples were genotyped using a genetic barcode and/or whole genome sequencing, followed by a genetic relatedness analysis.The study identified three key findings:(1) The parasite population continuously recombines, with no single genotype dominating, in contrast to viral populations;(2) The relatedness of parasites is influenced by both spatial and temporal distances; and(3) The lowest genetic relatedness among parasites occurs during the transition from low to high transmission seasons. The authors suggest that this latter finding reflects the increased recombination associated with sexual reproduction in mosquitoes.The results section is well-structured, and the figures are clear and self-explanatory. The methods are adequately described, providing a solid foundation for the findings. While there are no unexpected results, it is reassuring to see the anticipated outcomes supported by actual data. The conclusions are generally well-supported; however, the discussion on the burden of asymptomatic infections falls outside the scope of the data, as no specific analysis was conducted on this aspect and was not stated as part of the aims of the study. Nonetheless, the recommendation to target asymptomatic infections is logical and relevant.
**Recommendations for the authors:**

**Reviewer #1 (Recommendations for the authors):**
(1) The manuscript would benefit from additional detail in the methods section.a. Refer to Figure 1 when you describe the included studies and sample processing.

We added the reference to Figure 1 (Line 131).

b. While you describe each step in the pipeline, you do not specify the tools, packages, or environment used (the GitHub link is also non-functional). A graphic representation of the pipeline, with more bioinformatic details than Supplementary Figure S1, would be helpful. Add references to used tools and software created by others.

The GitHub link has been updated and is now functional. We find Figure S1 already heavy in details, adding in more would be detrimental to our will of it being an easily readable summary of our pipeline. Readers seeking in-depth explanation of our pipeline might be more interested in reading the methods section instead. We are very much committed to credit the authors of the tools that were essential for us to create our analysis pipeline. The two most relevant tools that we used are hmmIBD and the Fws calculation, which were both cited in the methods (Lines 148-152, 214-215).

c. What changed in the genotyping protocol after May 2016? Does it not lead to bias in the (temporal) analysis by leaving these loci in for samples collected before May 2016 and making them 'unknown' for the majority of samples collected after this date?

These 21 SNPs all clustered in 1 of the 4 multiplexes used for molecular genotyping, which likely failed to produce accurate base calls. We updated the text to include this information (Lines 198-200).

The rationale behind the discarding of these 21 SNPs for barcodes sampled after May 2016 was that they were consistently mismatching with the WGS SNPs, probably due to genotyping error as mentioned above. However, by replacing these unknown positions in the molecular barcodes with WGS SNPs, 141 samples did recover some of these 21 SNPs with the accurate base calls (Figure S3A). Additionally, we added an extra analysis to assess the agreement between barcodes and WGS data (Figure S3B).

d. Related to this, how are unknown and mixed genotypes treated in the binary matrix? How is the binary matrix coded? Is 0 the same as the reference allele? So all the missing and mixed are treated as references? How many missing and mixed alleles are there, how often does it occur and how does this impact the IBD analysis?

We acknowledge that the details that we provided regarding the IBD analysis were confusing. hmmIBD requires a matrix that contains positive or null integers for each different allele at a given loci (all our loci were bi-allelic, thus only 0 and 1 were used) and -1 for missing data. In our case, we set missing and mixed alleles to -1, which were then ignored during the IBD estimation. The corresponding text was updated accordingly (Lines 173-175).

e. By excluding households with less than 5 comparisons, are you not preselecting households with high numbers of cases, and therefore higher likelihood of transmission within the household?

All participants in each household were sampled at every collection time point. This sampling was unbiased towards likelihood of transmission. Excluding pairs of households with less than 5 comparisons was necessary to ensure statistical robustness in our analyses. Besides, this does not necessarily restrict the analysis to only households with a high number of cases as it is the total number of pairs between households that must equal 5 at least (for instance these pairs would pass the cutoff: household with 1 case vs household with 5 cases; household with 2 cases vs household with 3 cases).

(2) Since the authors only do the genotype replacement and build consensus barcode for 199 samples, there is a bias between the samples with consensus barcode and those with only the genotyping barcode. How did this impact the analysis?

See (6) in the Public Review.

a. It would be good to get a better sense of the distribution of the nr of SNPs in the barcode. The range is 30-89, and 30 SNPs for IBD is really not that much.

Adding the range of the number of available SNPs per barcode is indeed particularly relevant. We added a supplementary figure (Figure S5) showing the distribution of homozygous SNPs per barcode, showing that a very small minority of barcodes have only 30 SNPs available for IBD (average of 65, median of 64).

b. Did you compare the nr of SNPs in the consensus vs. only genotyped barcodes? Is there more missing data in the genotype-only barcodes?

We added a supplementary figure (Figure S5) with the distribution of homozygous SNPs in consensus (216 samples) and molecular (209 samples) barcodes. Consensus barcodes have more homozygous SNPs (average 76, median 82) than molecular barcodes (average of 54, median of 53), showing the improvement resulting from using whole genome sequencing data.

c. How was the cut-off/sample exclusion criteria of 30 SNPs in the barcode determined?

As described above (Public review section 7.a.), we removed pairs of barcodes with less than 30 comparable loci (and 10 informative loci) because this led to a good agreement between IBD values obtained from barcodes and genomes while still retaining a majority of pairwise IBD values.

d. Was there more/less IBD between sample pairs with a consensus barcode vs those with genotype-only barcodes?

We separated pairwise IBD values into two groups: “within consensus” and “within molecular”. The percentages of related barcodes (IBD ≥ 0.5) was virtually identical between “within consensus” (1.88 %) and “within molecular” (1.71 %) groups (χ^2^ = 1.33, p value > 0.24).

(3) Line 124 adds a reference for the PCR method used.

We have updated this information: varATS qPCR (Line 121).

(4) Line 126, what is MN2100ff? Is this the catalogue number of the cellulose columns? Please clarify and add manufacturer details.

MN2100ff was a replacement for CF11. We added a link to the MalariaGen website describing the product and the procedure (Lines 124-125).

(5) Line 143: Figure S7 is the first supplementary figure referenced. Change the order and make this Figure S1?

The numbering of figures is now fixed.

(6) Line 154: How many SNPs were in the vcf before filtering?

There were 1,042,186 SNPs before filtering. This information was added to the methods (Line 168).

(7) Line 156: Why is QUAL filtered at 10000? This seems extremely high. (I could be mistaken, but often QUAL above 50 or so is already fine, why discard everything below 10000?). What is the range of QUAL scores in your vcf?

We used the QUAL > 10000 to make our analyses less computationally intensive while keeping enough relevant genetic information. We agree that keeping variants with extremely high values of QUAL is not relevant above a certain threshold as it translates into infinitesimally low probabilities (10^-(QUAL/10)^) of the variant calling being wrong. We then decided to use a minimal population minor allele frequency (MAF) of 0.01 to keep a variant as this will make the IBD calculation more accurate (Taylor et al., 2019). The variant filtering was carried out with the MAF > 0.01 filter, resulting in 27,577 filtered SNPs with a minimal QUAL of 132. With a cutoff of 3000 available SNPs, we retrieved all 199 genomes previously obtained with the QUAL > 10000 condition. The methods have been updated accordingly (Lines 166-170).

(8) Line 161-165: How did you handle the mixed alleles in the hmmIBD analysis for the WGS data? Did you set them as 0 as you do later on for the consensus barcode?

Mixed alleles and missing data were ignored. This translated into a value of -1 for the hmmIBD matrix and not 0 as we incorrectly stated previously. We updated our manuscript with this correct information (Lines 173-175).

(9) Line 168-171: How many SNPs do you have in the WGS dataset after all the filtering steps? If the aim of the IBD with WGS was to validate the IBD-analysis with the barcode, wouldn't it make sense to have at least 200 loci (as shown in Taylor et al to be required for hmmIBD) in the WGS data? What proportion of comparisons were there with only 100 pairs of loci? This seems like really few SNPs from WGS data.

There were 27,577 SNPs overall in the 199 high quality genomes. In our analysis, we make the distinction between comparable and informative loci. For two loci to be comparable, they both have to be homozygous. To be informative, they must be comparable and at least one of them must correspond to the minor allele in the population. We borrowed this term and definition from hmmIBD software which yields directly the number of informative loci per pair. By keeping pairs with at least 100 informative SNPs, we aimed to reduce the number of samples artificially related because only population major alleles are being compared. Pairs of genomes had between 1073 and 27466 of these, way above the recommended 200 loci in Taylor et al. (2019). We added more details on comparable and informative sites (Lines 152-160).

(10) Line 178: why remove the 12 loci that are absent from the WGS? Are these loci also poorly genotyped in the spotmalaria panel?

As our goal is to validate the reliability of molecular genotyped SNPs, these 12 loci have to be removed. Especially because we did find a consistent discrepancy between genotyped and WGSed SNPs, which cannot be tested if these SNPs are absent from the genomes.

(11) Line 180-182: What do you mean by this sentence: "Genomic barcodes are built using different cutoffs of within-sample MAF and aligned against molecular barcodes from the same isolates." Is this the analysis presented in the supplementary figure and resulting in the cut-off of MAF 0.2? Please clarify.

A loci where both alleles are called can result from two distinct haploïd genomes present or from an error occurring during sequencing data acquisition or processing. To distinguish between the two, we empirically determined the cutoff of within-sample MAF above which the loci can be considered heterozygous and below which only the major allele is kept. The corresponding figure was indeed Figure S2 (referenced in next sentence Lines 192-195). We clarified our approach in the methods (Lines 190-192) and legends of Figures S2 and Figure S3.

(12) Line 191: How often was there a mismatch between WGS and SNP barcode?

We added a panel (Figure S3B) showing the average agreement of each SNP between molecular genotyping and WGS. We highlighted the 21 discrepant SNPs showing a lower agreement only for samples collected after May 2016.

(13) Line 201-204: This part is unclear (as above for the WGS): did you include sample pairs with more than 10 paired loci? But isn't 10 loci way too few to do IBD analysis?

We included pairs of samples with at least 30 comparable loci and 10 informative paired loci (refer to our answer to comment 8 for the difference between the two). We added more details regarding comparable and informative sites (Lines 152-160). Indeed, using fewer than 200 loci leads to an IBD estimation that is on average off by 0.1 or more (Taylor et al., 2019). However we showed that the barcode relatedness classification based on a cutoff of IBD (related when above 0.5, unrelated otherwise) was close enough to our gold standard using genomes (each pair having more than 1000 comparable sites). Because we use this classification approach rather than the exact value of barcode-estimated IBD in our study, our 30 minimum comparable sites cutoff seems sufficient.

(14) Lines 206-207: which program did you use to analyse Fws?

We did not use any program, we computed Fws according to Manske et al. (2012) methods.

(15) Line 233: "we attempted parasite genotyping and whole genome sequencing of 522 isolates over 16 time points" => This is confusing, you did not do WGS of 522 samples, only 199 as mentioned in the next sentence.

We attempted whole genome sequencing on 331 isolates and molecular genotyping on 442 isolates with 251 isolates common between the two methods. We updated our text to clarify this point (Lines 247-252).

(16) Lines 256-259: Add a range of proportions or some other summary statistic in this section as you are only referring here to supplementary figures to support these statements.

The text has been updated (Lines 271-274).

(17) Line 260: check the formatting of the reference "Collins22" as the rest of the document references are numbered.

Fixed.

(18) Figure 2/3:a. You could also inspect relatedness at the temporal level, by adjusting the network figure where the color is village and shape is time (month/year).

Although visualising the effect of time on the parasite relatedness network would be a valuable addition, we did not find any intuitive and simple way of doing so. Using shapes to represent time might end up being more confusing than helpful, especially because the sampling was not done at fixed intervals.

b. To further support the statement of clustering at the household level, it might be useful to add a (supplementary) figure with the network with household number/IDs as color or shape. In the network, there seems to be a lot of relatedness within the villages and between villages. Perhaps looking only at the distribution of the proportion of highly related isolates is simplifying the data too much. Besides, there is no statistical difference between clustering at the household vs within-village levels as indicated in Figure 3.

Unfortunately, there are too many households (71 in Figure 2) to make a figure with one color or shape per household readable. The statistical test of the difference between the within household and within village relatedness yielded a p value above the cutoff of 0.05 (p value of 0.084). However, it is possible that the lack of significance arises from the relatively low number of data points available in the “within household” group. This is even more plausible considering the statistical difference of both “within household” and “within village” groups with “between village” group. Overall, our results indicate a decreasing parasite relatedness with spatial distance, and that more investigation would be needed to quantify the difference between “within household” and “within village” groups.

(19) Figure 4: Please add more description in the caption of this figure to help interpret what is displayed here. Figure 4A is hard to interpret and does not seem to show more than is already shown in Figure 3A. What do the dots represent in Figure 4B? It is not clear what is presented here.

Compared to Figure 3A, Figure 4A enables the visualization of the relatedness between each individual pair of time points, which are later used in the comparison of relatedness between seasonal groups in Figure 4B. For this reason, we believe that Figure 4A should remain in the manuscript. However, we agree that the relationship between Figure 4A and Figure 4B is not intuitive in the way we presented it initially. For this reason, we added more details in the legend and modified Figure 4A to highlight the seasonal groups used in Figure 4B.

(20) Line 360-361: what did you do when haplotypes were not identical?

We explained it in the methods section (Lines 144-146): in this case, only WGS haplotypes were kept.

(21) Section chronic infections: it is important to mention that the majority of chronic infections are individuals from the monthly dry-season cohort.

We added a statement about the 21 chronically infected individuals that were also part of the December 2016 – May 2017 monthly follow-up (Lines 423-426).

(22) Lines 381-386: Did you investigate COI in these individuals? Could it be co-circulating strains that you do not pick up at all times due to the consensus barcodes and discarding of mixed genotypes (and does not necessarily show intra-host competition. That is speculation and should perhaps not be in the results)?

This is exactly what we think is happening. Due to the very nature of genotyping, only one strain may be observed at a time in the case of a co-infection, where distinct but related strains are simultaneously present in the host. The picked-up strain is typically the one with the highest relative abundance at the time of sampling. As the reviewer stated, fluctuation of strain abundance might not only be due to intra-host competition but also asynchronous development stages of the two strains. We added this observation to the manuscript (Lines 432-435).

(22) Figure 6: highlight the samples where the barcode was not available in a different color to be able to see the difference between a non-matching barcode and missing data.

We thank the reviewer for this great suggestion. We have now added to Figure 6 barcodes available along with their level of relatedness with the dominant genotypes for each continuous infections.

(24) Improve the discussion by adding a clear summary of the main findings and their implications, as well as study-specific limitations.

The Discussion has been updated with a paragraph summarizing the primary results (Lines 451-457).

(25) Line 445: "implying that the whole population had been replaced in just one year "a. What do you mean by replaced? Did other populations replace the existing populations? I am not sure the lack of IBD is enough to show that the population changed/was replaced. Perhaps it is more accurate to say that the same population evolved. Nevertheless, other measures such as genetic diversity and genetic differentiation or population structure.would be more suitable to strengthen these conclusions.

We agree that “replaced” was the wrong term in this case. We rather intended to describe how the numerous recombinations between malaria parasites completely reshaped the same initial population which gradually displayed lower levels of relatedness over time. We updated the manuscript accordingly (Lines 507-512).

**Reviewer #2 (Recommendations for the authors):**
(1) Line 260: Remove Collins 22.

Fixed.

(2) Lines 270-274: 73 + 213 = 286 not 284; sum of percentages is equal to 101%.

The numbers are correct: the 73 barcodes identical (IBD >= 0.9) to another barcode are a subset of the 213 related (IBD >= 0.5) to another barcode. However we agree that this might be confusing and will considering barcodes to be related if they have an IBD between 0.5 and 0.9, while excluding those with an IBD >= 0.9. The text has been updated (Lines 299-301).

(3) Section: "Independence of seasonality and drug resistance markers prevalence".

The text has been revised and the supplementary figure is now a main figure.

(4) For readers unaware of malaria control policy in the Gambia it would be helpful to have more details on the specifics of anti-malarial drug administration.

We added the drugs used in SMC (sulfadoxine-pyrimethamine and amodiaquine) and the first line antimalarial treatment in use in The Gambia during our study (Coartem) (Lines 383-388).

**Reviewer #3 (Recommendations for the authors):**
(1) The abstract is not as clear as the authors' summary. For example, I found the sentence starting with "with 425 *P. falciparum*..." hard to follow.

The abstract has been updated.

(2) It is better to consistently use "barcode genotyping "or "genotyping by barcode". Sometimes "molecular genotyping" is used instead of "barcode genotyping"

We have now replaced all occurrences of “barcode genotyping” with “molecular genotyping” or “molecular barcode genotyping”. We prefer to stick with “molecular genotyping” as this let us distinguish between the molecular and the genomic barcode.

(3) The introduction is quite disjoined and does not provide a clear build-up to the gap in knowledge that the study is attempting to fill. please revise.

Introduction is now thoroughly revised.

(4) Line 31 "with notable increase of parasite differentiation" is an interpretation and not an observation.

We have modified that sentence (Lines 31-33).

(5) Overall, the introduction requires substantial revision.

Introduction is now thoroughly revised.

(6) Line 70 "parasite population adapts..." I thought this required phenotypic analysis and not genetics?

The idea is that population of parasites may adapt to environmental conditions (such as seasonality) by selecting the most fitted genotypes. For instance, antimalarial exposure has an effect of selecting parasites with specific mutations in drug resistance related genes, and this even appears to be transient (for example with chloroquine). As such, there is good reason to think that seasonality might have a similar effect on parasite genetics.

(7) Line 129-130: the #442 is not reflected in the schematic Figure 1.

This is an intentional choice to make the figure more synthetic. For this reason, we included the Figure S1, which provides more details on the data collection and analysis pipeline.

(8) Line 242-243: "Made with natural earth". What is this?

This is a statement acknowledging the use of Natural Earth data to produce the map presented in Figure 1A.

(9) Line 260: "collins22", is this a reference?

Fixed.

(10) Line 269-70. Very hard to follow. Please revise.

We changed the text (Lines 293-297).

(11) Line 324: similarly... I think there is a typo here.

We did not find any typo in this specific sentence. However, “Similarly to Figure 3” sounds maybe a bit off, so we changed it to “As in Figure 3” (Line 351).

(12) Line 332-334: very hard to follow. please revise. Again, the lower parasite relatedness during the transition from low to high was linked to recombination occurring in the mosquito but what about infection burden shifting to naive young children? Is there a role for host immunity in the observed reduction in parasite-relatedness during the transition period?

This text has been rewritten (Lines 356-361).

About the hypothesis of infection burden shifting to naïve young children, this question is difficult to address in The Gambia because children under 5 years old received Seasonal Malaria Chemoprophylaxis during the high transmission season. In older children (6-15 years old), the prevalence was similar to adults (Fogang et al., 2024).

About the role of host immunity on parasite relatedness across time and space, our dataset is too small to divide it in different age groups. Further studies should address this very interesting question.